# Deep Learning-Based Near-Infrared Hyperspectral Imaging for Food Nutrition Estimation

**DOI:** 10.3390/foods12173145

**Published:** 2023-08-22

**Authors:** Tianhao Li, Wensong Wei, Shujuan Xing, Weiqing Min, Chunjiang Zhang, Shuqiang Jiang

**Affiliations:** 1The Key Laboratory of Intelligent Information Processing, Institute of Computing Technology, Chinese Academy of Sciences, Beijing 100190, China; 2University of Chinese Academy of Sciences, Beijing 100049, China; 3Institute of Food Science and Technology, Chinese Academy of Agricultural Sciences, Beijing 100193, China; 4Key Laboratory of Agro-Products Processing, Ministry of Agriculture and Rural Affairs, Beijing 100193, China

**Keywords:** deep learning, near-infrared hyperspectral imaging, food nutrition estimation, wavelength selection

## Abstract

The limited nutritional information provided by external food representations has constrained the further development of food nutrition estimation. Near-infrared hyperspectral imaging (NIR-HSI) technology can capture food chemical characteristics directly related to nutrition and is widely used in food science. However, conventional data analysis methods may lack the capability of modeling complex nonlinear relations between spectral information and nutrition content. Therefore, we initiated this study to explore the feasibility of integrating deep learning with NIR-HSI for food nutrition estimation. Inspired by reinforcement learning, we proposed OptmWave, an approach that can perform modeling and wavelength selection simultaneously. It achieved the highest accuracy on our constructed scrambled eggs with tomatoes dataset, with a determination coefficient of 0.9913 and a root mean square error (RMSE) of 0.3548. The interpretability of our selection results was confirmed through spectral analysis, validating the feasibility of deep learning-based NIR-HSI in food nutrition estimation.

## 1. Introduction

Diet plays a crucial role in maintaining health. Poor diet stands as a prominent instigator of diseases on a global scale, contributing to over a quarter of preventable fatalities across the world [1]. Food nutrition estimation is a way to ensure a healthy diet by allowing individuals to understand the specific nutritional value of foods [2]. Traditionally, it has relied on laboratory analysis, a time-consuming, costly, and labor-intensive process requiring specialized skills and equipment. While professional dietitians also offer a means of estimating food nutrition, their estimations may have slightly lower accuracy and may not fully align with the public’s increasing demand for convenient and reliable methods of daily food nutrition estimation.

In recent years, as a subdomain of food computing [3], food nutrition estimation techniques have emerged as promising alternatives. Most existing methods rely on food’s external representation to estimate its nutrition [4], including from the chewing sounds [5], food images [6,7,8], or ingredient text [9]. However, these methods are highly dependent on datasets and lack a comprehensive analysis of the chemical properties of the food, which is crucial for accurate nutrition estimation. A more promising approach is near-infrared hyperspectral imaging (NIR-HSI) which offers a direct and precise prediction of food nutrition by analyzing functional groups and molecular structures. NIR-HSI has been shown to be an effective tool for assessing various food quality parameters, including carbohydrate, fat, and protein composition [10,11]. Hu et al. leveraged off-the-shelf near-infrared emitting diodes and a photodiode to develop a portable calorie estimation system, demonstrating the great potential of near-infrared technology for food nutrition estimation in everyday life [12]. However, existing works in this field have mainly focused on simple foods such as staple foods, drinks, and fruits, which exhibit higher inter-class similarities in terms of their nutritional composition. Yet more complex food items, such as Chinese dishes, may exhibit larger variations in nutritional content due to differences in ingredient proportions. The accurate estimation of nutrition in such diverse food items remains a significant challenge. Therefore, to address this research gap and explore the potential of combining deep learning and NIR-HSI technology for food nutrition estimation, we present a case study on predicting the protein content in scrambled eggs with tomatoes. Protein plays a vital role in food nutrition as it provides essential amino acids and serves as a fundamental building block for human cells, tissues, and organs [13]. Consequently, accurately estimating the protein content in food is of great importance for nutritional assessment. In the context of Chinese dishes, scrambled eggs with tomatoes, a popular dish, has been selected as the focal point of this study. This choice is motivated by the dish’s simplicity, consisting of only two main ingredients, and the significant impact that ingredient proportions can have on its protein content. Therefore, analyzing the protein content in scrambled eggs with tomatoes serves as an illustrative case to explore the potential of combining deep learning and NIR-HSI technology for accurate food nutrition estimation.

The main contributions of this study are as follows: (1) Construct a near-infrared spectral dataset of scrambled eggs with tomatoes using meticulous measurements and processing. (2) Propose a novel deep learning method called OptmWave, which integrates two neural networks to simultaneously predict protein content and select wavelengths. (3) The comparison with conventional approaches demonstrates the effectiveness of our proposed deep learning methods. (4) The interpretability of our selection results is confirmed through near-infrared spectral analysis.

## 2. Material and Methods

### 2.1. Sample Preparation

In this study, tomatoes were purchased from Beijing Xingfurongyao supermarket, China, and eggs were bought from Beijing Hongyuan Technology Company, Beijing, China. The tomatoes were blanched and peeled using boiling water and then cut into 8 g chunks. Meanwhile, the eggs were evenly stirred using an eggbeater. The total weight of each sample was 560 g, consisting of raw materials (500 g) and seasonings (60 g). Based on the Chinese dietary guidelines, the seasonings used in the dish included oil (50 g), salt (5 g), and sugar (5 g). The dish incorporated varying proportions of eggs and tomatoes as raw materials: 50 g egg with 450 g tomatoes, 100 g egg with 400 g tomatoes, 150 g egg with 350 g tomatoes, 200 g egg with 300 g tomatoes, and 250 g egg with 250 g tomatoes. To diversify the sample types, the study also included samples containing only eggs or tomatoes.

The scrambled egg with tomatoes was prepared according to a traditional Chinese recipe: eggs and salt were evenly mixed, added to 160 °C oil, and fried for 1–1.5 min before the addition of tomatoes and sugar. The cooking time was 3.5 min, using a cooking power of 2100 W. Once cooled to an ambient temperature of 24 ${}^{\circ }C$, the samples were homogenized using a KENWOOD-AT320B cooking-chef machine at third gear for 30 s. The resulting mixture was poured into Petri dishes with a diameter of 3 cm and a height of 0.5 cm and filled to a weight of 8 g before being smoothed out.

### 2.2. Hyperspectral Image Acquisition and Spectra Extraction

#### 2.2.1. Hyperspectral Imaging System

The reflectance spectra of the scrambled eggs with tomatoes samples were recorded using a near-infrared (NIR) system provided by American Ocean Optics Company. The NIR system comprised a FLAME-S-VIS-NIR-ES spectrometer with a wavelength range of 900 to 1700 nm, an HL-2000-FHSA light source, a diffuse reflection standard plate, a reflection probe bracket, and a laboratory-level reflection probe with a diameter of 400 μm. For spectrum acquisition, OceanView 1.6.7 software was employed, which allowed for precise control and measurement of the reflectance spectra. The spectral resolution achieved with this setup was 3.1 nm, ensuring detailed characterization of the samples. The FLAME-S-VIS-NIR-ES spectrometer served as the primary instrument for capturing the reflectance spectra. It utilized the HL-2000-FHSA light source to illuminate the samples, and the reflected light was collected using the reflection probe. The probe, securely held in place by the reflection probe bracket, maintained consistent positioning during the measurements. The diffuse reflection standard plate was employed as a reference to calibrate the system and ensure accurate measurements. The process of hyperspectral image acquisition is illustrated in the Figure 1.

#### 2.2.2. Hyperspectral Image Correction

To minimize the impact of solar altitude angle variations on spectral measurements, sample spectra were obtained between 10:00 and 14:00 Beijing time. Throughout this time frame, the solar altitude angle remained above 45 degrees [14]. Prior to data collection, both the light source and spectrometer were preheated for 30 min, and the sample was maintained at room temperature (24 degrees Celsius). After cooling, the following parameters were set for collecting spectra of scrambled eggs with tomatoes: an integration time of 196 ms, 10 scanning repetitions, and 5 smoothing points. The spectral data were captured within the range of 900 to 1700 nm, with a spectral interval of approximately 3.1 nm. A diffuse reflectance standard served as the white reference, while the dark current was recorded by deactivating the light source. Furthermore, to achieve black–white calibration, the following formula was employed:(1)$$R_{\mathrm{calibrated}}=\frac{R_{\mathrm{measured}}-R_{\mathrm{dark}}}{R_{\mathrm{white}}-R_{\mathrm{dark}}}$$
where $R_{\mathrm{calibrated}}$ represents the calibrated hyperspectral image, $R_{\mathrm{measured}}$ represents the measured hyperspectral image, $R_{\mathrm{dark}}$ corresponds to the dark current hyperspectral image, and $R_{\mathrm{white}}$ corresponds to the white reference intensity.

#### 2.2.3. Spectra Extraction

After correction, the spectra of each sample were extracted by utilizing the default band parameters of the instrument (1453.51 nm, 1246.54 nm, 1123.34 nm) as the RGB channels for hyperspectral image visualization, followed by the application of the Hough transform for sample position detection. This computer vision technique, commonly used for object detection in images, works by transforming the image into a parameter space to identify peaks corresponding to the presence of a particular shape. The detected position information was then utilized to automatically delineate the region of interest (ROI) for each sample. Finally, the sample’s spectral information was extracted by averaging the data within the ROI. The process of spectra extraction is illustrated in Figure 1.

### 2.3. Reference Measurement of Protein Content

The Kjeldahl method was used, as specified in GB 5009.5-2016, for measuring the protein content of the scrambled eggs with tomatoes. A thoroughly mixed sample weighing 0.1 g was added to 0.4 g of copper sulphate, 6 g of potassium sulphate, and 20 mL of sulphuric acid in a digestion furnace for digestion. After digestion, the green and transparent liquid in the digestive tube was cooled and automatically dosed, distilled, titrated, and recorded using an automatic Kjeldahl nitrogen determinator (FOSS-Kjeltec 2300). The protein content of the sample was calculated using the following equation:(2)$$Z=\frac{(V_{1}-V_{2})\times c\times 0.014}{m\times V_{3}/100}$$
where Z represents the protein content of the sample (g/100 g); $V_{1}$ is the volume of the reagent blank that consumes the hydrochloric acid standard titration solution (mL); $V_{2}$ is the volume of the standard titration solution of sulphuric acid or hydrochloric acid consumed by the reagent blank (mL); *c* is the concentration of the hydrochloric acid standard titration solution (mol/L); 0.014 is the mass of nitrogen equivalent to hydrochloric acid [c(HCl) = 1.000 mol/L] standard titration solution (g); *m* is the mass of the sample (g); $V_{3}$ is the volume of absorbed digestive juice (mL); and *F* is the reduction coefficient (6.25), where 100 is the conversion factor.

### 2.4. A Deep Learning Framework for Wavelength Selection and Regression Simultaneously

#### 2.4.1. Overall Network Architecture

To gain deeper insights into the correlation between the wavelength and protein content, we present the OptmWave framework, a novel approach for effective wavelength selection in spectral data. Inspired by INVASE [15], OptmWave integrates two neural networks: the Selection Probability Generation Network (SPGN) and the Prediction Network (PN). This framework addresses the challenge of accurately identifying the most informative wavelengths and predicting protein content. By leveraging the SPGN and PN, OptmWave streamlines the wavelength selection process and enhances protein content predictions in a spectral data analysis. The architecture of our framework is illustrated in Figure 2a.

Our framework also inherits the general design of INVASE. The SPGN could be any neural network with the same input and output dimensions, while the PN could be any regression network. However, considering the specific characteristics of the dataset in this case study, it is still necessary to perform a detailed design for the SPGN and PN. A Multilayer Perceptron (MLP) is chosen as the foundational structure. An MLP is a type of artificial neural network that consists of an input layer, one or more hidden layers, and an output layer. The input layer receives the input data, and the hidden layers perform the computation to extract features and representations of the input data. The output layer produces the final prediction result. The principle of an MLP is based on the concept of artificial neurons, which are modeled after the biological neurons in the human brain. Each neuron receives inputs from the previous layer and produces an output signal by applying a nonlinear activation function. The weights and biases of the neurons are learned through the training process to minimize the prediction error.

The detailed design of the SPGN adopted in this work is illustrated in Figure 2b. It consists of an MLP with three hidden layers, incorporating dropout layers in both the input and hidden layers to enhance its robustness. The output layer is connected to a sigmoid function to ensure that the output values are bounded between 0 and 1. The activation function is the Scaled Exponential Linear Unit (SELU), which is a self-normalizing activation function that ensures that the activations of each layer remain close to the mean and variance of the inputs, regardless of the network’s depth, stabilizing the training process and leading to faster convergence [16].

The detailed design of the PN adopted in this work is illustrated in Figure 2c. The PN has three hidden layers of size 128, 64, and 32, respectively, with layer normalization (LN) [17] layers connected after the input and hidden layers. LN is a technique that normalizes the activations of each neuron across the features dimension, rather than the batch dimension, like in batch normalization. This technique has been shown to be effective in stabilizing the training process and improving the generalization performance of deep neural networks. The activation function used is the SELU. It is worth noting that the PN can be trained as an MLP regression model independently to predict protein content. Therefore, it can be combined with conventional wavelength selection methods. Consequently, in the experimental section, we will also evaluate the design of this part separately.

#### 2.4.2. Joint Training Strategy

OptmWave includes our proposed joint training strategy similar to the critic–actor architecture in reinforcement learning that optimizes both neural networks. The SPGN operates by receiving a full-spectrum input vector x of dimension d. This input vector represents the spectral data, with each dimension corresponding to a different wavelength. Then, it generates an output vector p of the same dimension, with each value of p representing a probability between 0 and 1, indicating the likelihood of selecting each corresponding wavelength in the input vector. The probability vector p is used for Bernoulli sampling to obtain a selection result vector s of dimension d, consisting of only 0 s and 1 s. A value of 0 signifies that the corresponding wavelength is not selected, while a value of 1 denotes its selection. The selection result vector s is element-wise multiplied with the input vector x, yielding the selected spectral data $x^{*}$.

The PN receives the selected spectral data $x^{*}$ and performs one iteration of training. To estimate the performance of the current selection result s, the trained PN is then evaluated on $x^{*}$. The prediction result is used with the true values to calculate the $R^{2}$ score as an evaluation metric. The $R^{2}$ score measures the goodness of fit between the predicted and true values and is calculated as 1 min the ratio of the sum of squared errors of the predicted values to the sum of squared errors of the mean of the true values. A reward is defined to quantify the performance of the SPGN:(3)$$\mathrm{Reward}=\alpha (R^{2}-\varphi )$$
where $R^{2}$ represents the $R^{2}$ score on the validation set, φ denotes a baseline $R^{2}$ score that serves as a reference point for comparison, and α is a scaling factor employed to adjust the $R^{2}$ score to a suitable range. The reward can be either positive or negative, depending on the performance of the PN after training with the selected spectral data $x^{*}$. A positive reward indicates that s is a good selection result, as it leads to favorable prediction performance on the training set. In contrast, a negative reward implies that the selection result may not be suitable and should be avoided for future iterations. Then, the SPGN can perform one iteration of training according to the reward.

After each iteration of joint training, where the SPGN and PN are trained according to the proposed strategy, we evaluate the OptmWave’s performance on the validation set. The training process continues until the SPGN converges, but there is no guarantee that the PN will also converge. Therefore, we fix the SPGN’s weight and train the PN separately. At the start, we train the PN with the selected spectral data obtained from the SPGN in a single iteration. After this training, we evaluate the PN’s performance on the validation set. If the PN reaches convergence, we stop training and select the optimal model based on its performance on the validation set. If the PN does not reach convergence, we continue training with the same spectral data from the SPGN until we achieve satisfactory performance. When both the SPGN and PN converge, they can be taken out and used separately. The SPGN prints the same mask, representing the final selection result. The PN can predict the protein content using spectral data after the wavelength selection.

#### 2.4.3. Loss Functions

To successfully implement the proposed joint training strategy, well-designed loss functions are essential. In the case of OptmWave, where the PN functions as a regression model, we have opted for the mean squared error (MSE) as the chosen loss function. Widely used for regression problems, the MSE calculates the average squared difference between predicted and actual values, providing a suitable measure for training the PN.

Regarding the SPGN loss function, denoted as $L_{\mathrm{SPGN}}$, it comprises two main components: $L_{\mathrm{Selection}}$ and $L_{\mathrm{Sparsity}}$. These components are effectively balanced through the hyperparameter λ. During the training process, the combined loss function is utilized to update the SPGN, taking into consideration the associated reward. By incorporating both selection and sparsity considerations, this approach ensures the effectiveness and efficiency of the SPGN’s training process. The loss function of the SPGN is defined as follows:(4)$$L_{\mathrm{SPGN}}=\left(L_{\mathrm{Selection}}+\lambda L_{\mathrm{Sparsity}}\right)\times \mathrm{Reward}\;$$
(5)$$L_{\mathrm{Selection}}=-\sum_{i=1}^{d}s_{i}\log\left(p_{i}\right)+\left(1-s_{i}\right)\log\left(1-p_{i}\right)$$
(6)$$L_{\mathrm{Sparsity}}=\sum_{i=1}^{d}p_{i}\;$$

$L_{\mathrm{Selection}}$ quantifies the discrepancy between the selection result vector s and the probability vector p. It is calculated via the cross-entropy loss function and is thus suitable for optimization problems involving probabilities. A smaller value signifies a better match between the selection policy and the actual wavelength selection, as the probability vector p is closer to the selection result vector s. $L_{\mathrm{Sparsity}}$ encourages wavelength selection sparsity by penalizing the sum of probabilities in the probability vector p. A smaller value corresponds to a sparser selection, reducing the number of selected wavelengths for more efficient computations and better generalization performance. The hyperparameter λ balances the contributions of these two loss components, controlling the trade-off between matching the selection policy to the actual wavelength selection and promoting sparsity in the selection process. The optimization process for $L_{\mathrm{SPGN}}$ is influenced by the reward, which depends on the performance of the PN on the training set after training with the selected spectral data s. A positive reward signifies that the current wavelength selection results in good prediction performance. The optimizer minimizes $L_{\mathrm{SPGN}}$ to encourage the SPGN to maintain or improve the wavelength selection policy. This update enhances the match between the probability vector p and the selection result vector s while promoting wavelength sparsity. A negative reward suggests that the current wavelength selection leads to poor performance. The optimizer maximizes $L_{\mathrm{SPGN}}$, guiding the SPGN to update its weights to avoid the detrimental wavelength selection policy and explore alternative strategies by reducing sparsity in the selected wavelengths. The optimizer adjusts the SPGN’s weights based on the reward sign, aiming to minimize $L_{\mathrm{SPGN}}$ when the reward is positive and maximize it when the reward is negative. This adaptive learning process refines the SPGN’s wavelength selection policy to achieve better prediction performance and sparsity. The overall loss function, $L_{\mathrm{SPGN}}$, is obtained by adding the weighted selection and sparsity loss components and then multiplying the sum by the reward. This ensures that the SPGN is updated based on its performance, allowing it to learn an effective wavelength selection policy while considering both prediction performance and the sparsity of the selected wavelengths.

### 2.5. Conventional Approaches

#### 2.5.1. Common Regression Models

In the field of near-infrared (NIR) spectroscopy analysis, there are several conventional data analysis approaches that are commonly used for quantitative prediction of the physical and chemical properties of samples based on their NIR spectra. Two of the most common regression models in NIR analysis are partial least squares (PLS) [18] and support vector regression (SVR) [19]. PLS is a linear regression technique that seeks to model the relationship between the NIR spectra and the property of interest by constructing a set of orthogonal latent variables, which are also known as latent factors. The latent variables are constructed in such a way so as to maximize the covariance between the NIR spectra and the property of interest. The regression model is then built based on the latent variables instead of the original NIR spectra. SVR is a type of machine learning algorithm that can be used for regression problems. It aims to find the optimal boundary in the feature space that separates the samples into two classes and then maps the samples to a high-dimensional feature space where a linear regression model can be applied. The regression model is built based on a subset of the training samples, which are known as support vectors, and the optimization of the model is performed by minimizing the distance between the support vectors and the boundary.

#### 2.5.2. Effective Wavelength Selection

Efficient wavelength selection constitutes a crucial phase in NIR data analysis, given the potential hindrance posed by high-dimensional, uninformative, and redundant variables to the precise interpretation of wavelength-related information. There are several effective wavelength selection strategies, such as the successive projections algorithm (SPA) and competitive adaptive reweighted sampling (CARS). The SPA operates by projecting one variable onto the others to identify candidate wavelengths. In combination with the weighted regression coefficient analysis, the SPA assists in determining the optimal wavelengths for use in multivariate regression models [20]. One PLS model is created via Monte-Carlo sampling in the CARS. The CARS method involves creating a PLS model via Monte-Carlo sampling. This process produces regression coefficient magnitudes [21], which are associated with each wavelength. A decaying exponential function is then used to exclude wavelengths with the smallest magnitudes. By employing adaptive reweighted sampling, candidate subsets can be derived. After that, the subset that aligns with the PLS models showing the lowest RMSE during cross-validation is considered as the optimal wavelength set.

### 2.6. Model Evaluation

All the procedures described were implemented using Python. The Spectral Python library was employed for processing hyperspectral imaging data, while OpenCV and NumPy were utilized for spectra extraction. For conventional data analysis approaches, sklearn was used, whereas PyTorch was employed for deep learning approaches.

The performances of the models were assessed using three evaluation metrics: coefficient of determination ($R^{2}$), root mean square error (RMSE), and predictive residual deviation (RPD). These metrics are defined as follows: $R^{2}$ measures the proportion of the variance in the dependent variable that can be explained by the independent variable(s). It provides an indication of how well the model fits the observed data, with values ranging from 0 to 1. A higher $R^{2}$ value indicates a better fit. The formula for $R^{2}$ is given by
(7)$$R^{2}=1-\frac{\sum_{i=1}^{n}{\left(y_{i}-\hat{y_{i}}\right)}^{2}}{\sum_{i=1}^{n}{\left(y_{i}-\bar{y}\right)}^{2}}$$
where $y_{i}$ represents the observed values of the dependent variable, $\hat{y_{i}}$ represents the predicted values of the dependent variable, $\bar{y}$ represents the mean of the observed values of the dependent variable, and *n* represents the number of data points.

The RMSE measures the average deviation between the predicted values and the observed values. The RMSE provides an estimate of the model’s accuracy, with lower values indicating better performance. The formula for RMSE is given by
(8)$$\mathrm{RMSE}=\sqrt{\frac{\sum_{i=1}^{n}{\left(y_{i}-\hat{y_{i}}\right)}^{2}}{n}}$$
where $y_{i}$ represents the observed values of the dependent variable, $\hat{y_{i}}$ represents the predicted values of the dependent variable, and *n* represents the number of data points.

The RPD is a measure of the ratio between the standard deviation of the reference values and the standard error of prediction. The RPD is commonly used in spectroscopy to evaluate the precision of models. Higher RPD values indicate better predictive performance. The formula for RPD is given by
(9)$$\mathrm{RPD}=\frac{\mathrm{Standard}\;\mathrm{Deviation}\;\mathrm{of}\;\mathrm{Reference}\;\mathrm{Values}}{\mathrm{Standard}\;\mathrm{Error}\;\mathrm{of}\;\mathrm{Prediction}}$$

The optimal model is determined based on the model that achieves the highest $R^{2}$ and RPD values while also minimizing the RMSE value.

## 3. Results

### 3.1. Spectral Profiles

The preprocessed reflectance spectra of the samples are illustrated in Figure 3a. To eliminate obvious noise at the head and end of the spectra, only the range of 900–1700 nm was investigated. As shown in Figure 3b, the spectral data for all samples demonstrated a consistent pattern of peaks and valleys. The major absorptions were observed at valleys around 970, 1206, and 1440 nm. The local absorption maxima at 970 nm and 1440 nm (O-H stretching second and first overtones, respectively) were attributed to the presence of water in the sample [22]. Also, the valley around 1206 nm, which corresponds to the second overtone of C-H stretching, can be attributed to the fat content of the sample [23,24].

### 3.2. Sample Set Split

A total of 487 distinct samples were collected and measured. Before partitioning the dataset, spectral anomalies were detected using the Hotelling $T^{2}$ test and subsequently removed. The remaining samples were randomly shuffled and divided into training, validation, and test sets with a ratio of 6:2:2. The final training, validation, and test sets consisted of 292, 98, and 97 samples, respectively. Table 1 presents the minimum value, maximum value, mean, and standard deviation of the protein content for each data split.

### 3.3. Experimental Setup

#### 3.3.1. Deep Learning Approaches

For the case of using the PN only, the learning rate was set to $10^{-4}$. And for the OptmWave, the hyperparameters were set as follows: the learning rate of the SPGN was $5\times 10^{-5}$, the learning rate of the PN was $2\times 10^{-3}$, α was 100, φ was 0.95, and λ was 0.1. The hyperparameters were manually tuned based on the performance on the validation set, and φ was set based on the prediction performance of the PLS. Compared to the case of using the PN only, the joint training strategy requires the PN to adapt to continuously changing training data, thus necessitating a larger learning rate.

#### 3.3.2. Conventional Approaches

In this study, we utilized PLS and SVR as conventional prediction models and employed the SPA and CARS methods to select effective wavelengths. To optimize the performance of each model, we employed different strategies. For the PLS model, we utilized a grid search approach. To elaborate, a grid search strategy incorporating a 5-fold cross-validation was utilized. The grid search is a brute-force approach that exhaustively searches over a predefined parameter space to find the optimal set of hyperparameters. The search parameters involved were the maximum number of iterations with values of 500, 1000, and 2000; the tolerance with values of $10^{-4}$, $10^{-3}$, and $10^{-2}$; and the number of components ranging from two to the specified maximum number of components. For the SVR model, we used an automated approach called Autosklearn [25] to search for the optimal hyperparameters. Autosklearn is an automated machine learning tool that applies Bayesian optimization to find the optimal hyperparameters for a given dataset. The search time was set to two hours to find the best SVR hyperparameters.

### 3.4. Result Analysis

The experimental results are summarized in Table 2. For the conventional approaches, the PLS and SVR models using full-spectra and selected wavelengths all obtained good results. The R${}^{2}$ score of all the models is over 0.9, and the RPD is over 2.5. The comparison of the SVR and PLS models using the same full-spectra and wavelength selection dataset revealed that the SVR outperformed the PLS in the training, validation, and testing sets, possibly due to SVR’s ability to model nonlinear relationships in the data.

Among the PLS models, the CARS method yielded the best prediction accuracy of 0.9558. However, the results of applying CARS to the SVR model were not as satisfactory, possibly due to the algorithm’s coupling with PLS. In contrast, the SPA showed slightly better results than using the full-spectra data, likely because of its ability to reduce multicollinearity in the data.

And for the case of using the PN only, the experimental results showed that the accuracy on both the full-spectrum data and the wavelength selection data was higher than that of the conventional models. This is because the deep learning model can capture more complex nonlinear relationships in the data, as demonstrated by its higher accuracy on the training set. However, this also leads to a decrease in accuracy on the wavelength selection data, as the reduction in wavelength selection reduces the amount of information available for prediction. The CARS results were higher than those of the PLS, possibly because CARS selected more wavelengths.

OptmWave reached the highest prediction performance. Figure 3c shows the visualization of the reference versus predicted values for the protein content from the OptmWave. It can be seen that the protein content can be well-predicted for all samples in the test set. The experiments were also designed to assess the efficacy of the wavelength selection outcomes between OptmWave, CARS, and the SPA. Figure 3d shows the visualization of the effective wavelength selection result by the SPA, CARS, and our method SPGN. In detail, 917.4, 924.03, 943.93, 950.57, 967.16, 970.48, 987.08, 1010.32, 1050.19, 1063.48, 1183.25, 1189.91, 1193.24, 1199.9, 1203.23, 1206.56, 1209.89, 1216.55, 1243.21, 1276.55, 1286.55, 1296.56, 1323.25, and 1346.61 nm were selected. From the experimental results in Table 2, compared with the wavelengths selected by the SPA and CARS, our method achieved competitive results: on the PLS model, the prediction performance was close to that of the SPA and higher than that of the full-spectrum data; the prediction performance of the SVR model was better than CARS and the number of selected wavelengths was less than CARS. Good results were achieved even if the prediction was made directly using the selected wavelengths from the SPGN without using the joint training strategy.

We also try to explain the selection result by the vibration of chemical bonds. The selected wavelengths at 970.48, 987.08, and 1010.32 nm correspond to the N-H stretch second overtone locations associated with proteins [26]. In addition, wavelengths at 1276.55, 1286.55, 1296.56, 1323.25, and 1346.61 nm are also associated with proteins. They can be attributed to a combination of the first overtone of a specific N-H vibration (in Fermi resonance with N-H in-plane bend) with the fundamental vibrations of the N-H in-plane bend and C-N stretch with the N-H in-plane bend [27]. Other wavelengths selected may be indirectly related to the protein content. For example, the wavelengths around 1200 nm (1183.25, 1189.91, 1193.24, 1199.9, 1203.23, 1206.56, 1209.89, and 1216.55 nm) correspond to the C-H stretching second overtone [22,27,28]; 1363.3 nm is related to the C-H combination second overtone [29]; and 917.4 nm and 924.03 nm correspond to the C-H stretching third overtone from methyl and methylene, respectively [26]. And these wavelengths are associated with aliphatic compounds. From 950 nm to 1050 nm, the wavelengths at 943.43, 950.57, and 967.16 nm are related to the third overtone of C-H and the third overtone region of O-H from oil nutrient [30,31,32]. In this study, it has been observed that the variation among the samples is primarily attributed to the proportion of tomato and egg, which display significant differences in their aliphatic compounds and oil content. It is possible that deep learning models could use the spectra associated with these compounds to uncover their hidden relationship with protein content. Furthermore, our method does not include wavelengths beyond 1400 nm, as shown in Figure 3b,d, where reflectance is low and variance is small within the range of 1400 nm to 1700 nm. Protein information around 1500 nm, specifically the N-H stretching first overtone, may be masked by water information [22]. This limited ability to discriminate may have resulted in the exclusion of this spectral region in our method selection.

### 3.5. Ablation Study on Deep Learning Approaches

Ablation studies are conducted on deep learning approaches to evaluate the effectiveness of specific design choices. The experimental results for this part are summarized in Table 3. For the case of using the PN only, the performance without using the SELU, LN, or both was tested. In the absence of the SELU, a replacement was carried out using the ReLU activation function, while the omission of all the layer normalization layers was implemented in the absence of LN. This led to observed declines in varying degrees in the prediction results, thus affirming the efficacy of these designs.

Additionally, the performance of OptmWave without the proposed joint training strategy was evaluated. This involved training the PN directly using the selected wavelength data from the SPGN in Table 2. The reduction in prediction accuracy highlighted the importance of the joint training strategy. Our approach enables the MLP to gradually adapt to changes from the full-spectrum training data to the selected wavelengths, which has an effect similar to fine-tuning on a pre-trained model. The performance of OptmWave was also evaluated without our designed $L_{\mathrm{SPGN}}$. Instead, the loss function design from the implementation of INVASE was adopted [15], where $L_{\mathrm{Sparsity}}$ is not affected by the value of the reward. This resulted in poor results because the optimizer continuously tried to reduce the number of selected wavelengths without considering the prediction performance. That is mainly because INVASE is a method used for classification tasks to obtain the importance of input variables at the instance level. In contrast, our method is used for global variable selection on regression tasks.

## 4. Discussion

In the present study, the feasibility of deep learning-based NIR-HSI methods in predicting the protein content of specific foods is confirmed. This confirmation was established through a comparison of the accuracy and interpretability of various conventional methods and near-infrared spectroscopy analysis. The results demonstrate the significant potential of deep learning-based NIR-HSI in predicting food nutrition.

Establishing and analyzing models in the field of NIR-HSI holds significant importance [33]. According to Table 2, conventional methods have displayed underfitting in the training set: the models established could not fully fit the training set data. This is typically due to insufficient model complexity. Given identical data distribution, the better the results on the training set, the better they will perform on the test set. This is presumably the primary reason why deep learning methods outperformed conventional methods on our constructed dataset. Additionally, the SPGN and PN are not limited to our proposed design. For instance, the SPGN could be any neural network with the same input and output dimensions, while the PN could be any regression network. This design offers the possibility of extending to other tasks in food science. Researchers can design an SPGN and a PN according to their needs, and use the OptmWave method for training, thus obtaining models suitable for their tasks. We believe that this will promote the cross-disciplinary development of food science and artificial intelligence to some extent.

Despite the satisfactory results, our method does have some limitations. Firstly, deep learning methods require a certain amount of data. With a smaller dataset, the results might be comparable to, or even worse than, conventional methods [24]. Secondly, data distribution should be consistent, especially when partitioning into training, validation, and test sets. If there is a significant difference in distribution, it could lead to severe overfitting. Finally, the sampling mechanism in OptmWave increases the randomness of the method. If the random seed is not fixed, the results may vary.

Based on our findings, future research could further explore new paradigms in food nutrition assessment. For example, combining deep learning models capable of recognizing foods [34] with NIR-HSI methods for specific foods to accurately predict food nutrition. Furthermore, future research could focus on creating a large-scale benchmark dataset for food spectra [35]. Deep learning methods could then be applied to various foods, thereby establishing a large model for food spectra, empowering various downstream tasks in food science.

## 5. Conclusions

In this study, the feasibility of using deep learning approaches in determining the protein content of a specified food using near-infrared hyperspectral imaging is explored. A framework is proposed to achieve better prediction performance and effective wavelength selection. It is compared to conventional prediction approaches, such as PLS and SVR, and effective wavelength selection approaches, such as the SPA and CARS. The results demonstrated the effectiveness of deep learning approaches and showed that our proposed framework and wavelength selection method outperformed the conventional approaches. Our study highlights the great potential of deep learning-based near-infrared hyperspectral imaging in predicting nutrient content in food. The findings support the application of deep learning methods in food science and offer new avenues for food quality control and nutrition monitoring. Considering the practical implications of our research, the application of deep learning techniques can lead to increased automation and reduced human involvement. This could contribute to the development of more portable and affordable devices for food assessment. Looking ahead, our future studies will focus on investigating the application of these deep learning approaches in assessing nutrient content in a wider array of foods. The goal is to further enhance the prediction performance and broaden the range of practical applications.

## Figures and Tables

**Figure 1 foods-12-03145-f001:**
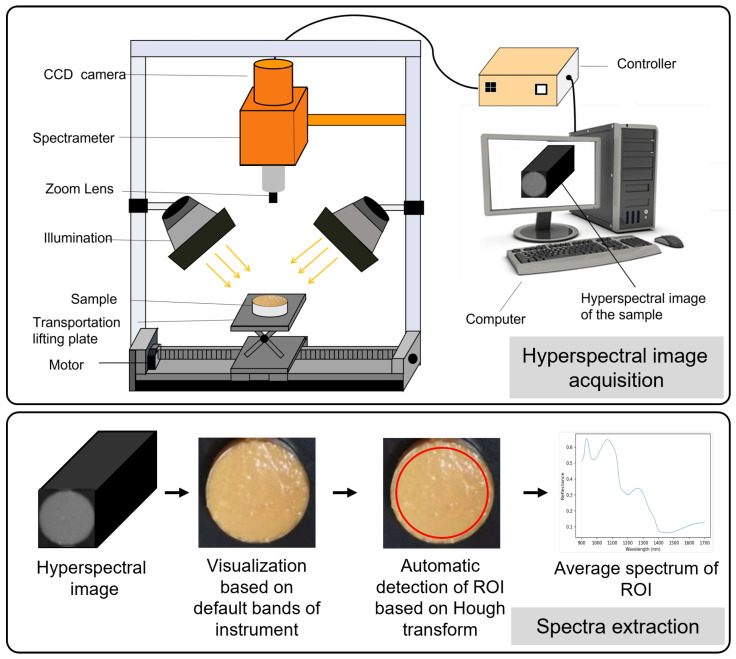
The process of hyperspectral image acquisition spectra extraction.

**Figure 2 foods-12-03145-f002:**
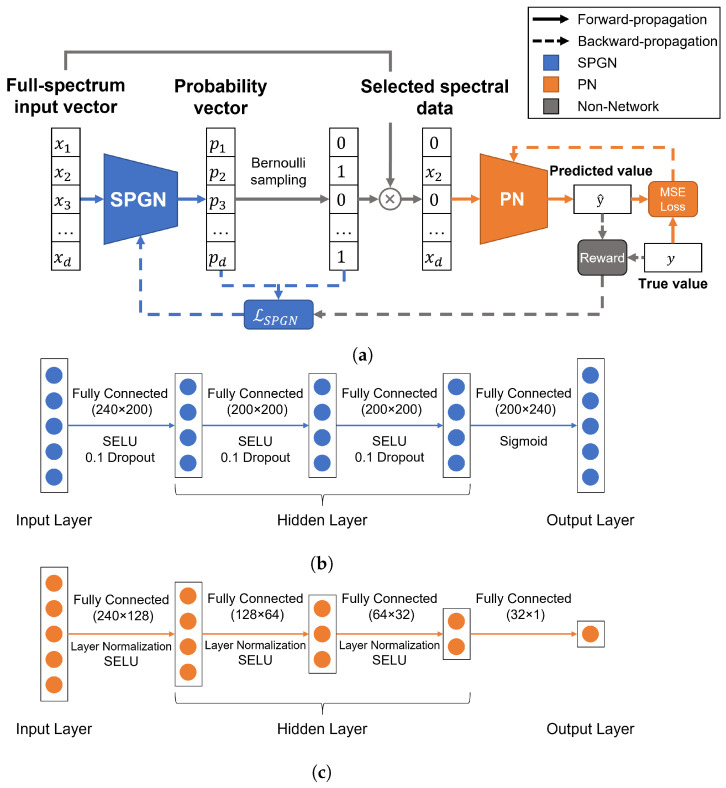
(**a**) The architecture of our proposed OptmWave framework. (**b**) Detailed design of SPGN. (**c**) Detailed design of PN.

**Figure 3 foods-12-03145-f003:**
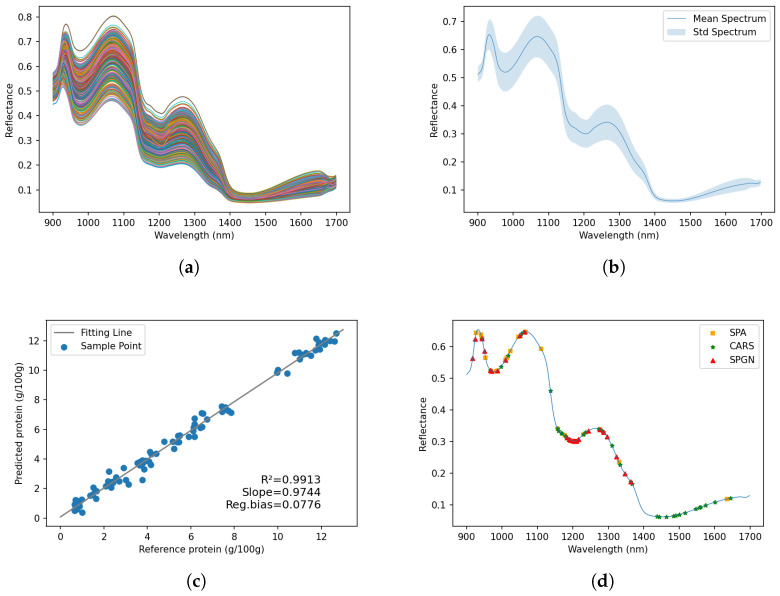
Spectral characterization for the dataset and the visualization of experiment results. (**a**) Reflectance spectra of all samples. (**b**) Mean spectrum and standard deviation of the spectral data. The blue line shows the mean spectrum, while the shaded area represents the standard deviation of the reflectance values at each wavelength. (**c**) Reference versus predicted values for protein content from OptmWave. (**d**) Visualization of effective wavelength selection result by SPA, CARS, and our proposed SPGN.

**Table 1 foods-12-03145-t001:** The statistical summaries of the protein content for each data split.

Data Split	Minimum (g/100 g)	Maximum (g/100 g)	Mean (g/100 g)	Standard Deviation (g/100 g)
Training set	0.6258	12.9541	5.4654	3.7056
Validation set	0.6016	13.1858	5.5092	3.6868
Test set	0.6505	12.6843	5.3266	3.8062

**Table 2 foods-12-03145-t002:** Prediction results of protein content.

Models	Data Type	N.V. *	Training Set		Validation Set			Test Set		
			R^2^	RMSE	R^2^	**RMSE**	RPD	R^2^	RMSE	RPD
PLS	Full	240	0.9624	0.7187	0.9689	0.6497	5.6742	0.9487	0.8620	4.4154
	CARS	37	0.9673	0.6697	0.9745	0.5887	6.2628	0.9558	0.8004	4.7553
	SPA	19	0.9535	0.7990	0.9638	0.7015	5.2556	0.9507	0.8449	4.5051
	SPGN	25	0.9569	0.7691	0.9645	0.6943	5.3104	0.9504	0.8478	4.4897
SVR	Full	240	0.9872	0.4189	0.9833	0.4767	7.7335	0.9772	0.5748	6.6219
	CARS	37	0.9799	0.5247	0.9781	0.5453	6.7605	0.9649	0.7131	5.3372
	SPA	19	0.9835	0.4762	0.9807	0.5116	7.2059	0.9773	0.5740	6.6314
	SPGN	25	0.9722	0.6182	0.9749	0.5845	6.3075	0.9653	0.7090	5.3684
PN	Full	240	0.9922	0.3267	0.9834	0.4748	7.7644	0.9876	0.4238	8.9802
	CARS	37	0.9926	0.3191	0.9814	0.5027	7.3334	0.9822	0.5073	7.5021
	SPA	19	0.9814	0.5061	0.9720	0.6167	5.9779	0.9754	0.5971	6.3740
	SPGN	25	0.9945	0.2761	0.9840	0.4666	7.9021	0.9852	0.4633	8.2159
OptmWave		25	0.9938	0.2920	0.9796	0.5263	7.0045	0.9913	0.3548	10.7278

* N.V. is the number of variables.

**Table 3 foods-12-03145-t003:** The results of ablation study on proposed deep learning approaches.

Method	Training Set		Validation Set			Test Set		
	R^2^	**RMSE**	R^2^	RMSE	RPD	R^2^	RMSE	RPD
PN	0.9922	0.3267	0.9834	0.4748	7.7644	0.9876	0.4238	8.9802
w/o SELU	0.9902	0.3667	0.9811	0.5075	7.2643	0.9773	0.5731	6.6409
w/o LN	0.9566	0.7717	0.9564	0.7694	4.7917	0.9574	0.7859	4.8428
w/o LN and SELU	0.9475	0.8488	0.9599	0.7379	4.9965	0.9460	0.8845	4.3032
OptmWave	0.9938	0.2920	0.9796	0.5263	7.0045	0.9913	0.3548	10.7278
w/o joint training strategy	0.9945	0.2761	0.9840	0.4666	7.9021	0.9852	0.4633	8.2159
w/o $L_{\mathrm{SPGN}}$	0.9363	0.9349	0.8011	1.6444	2.2420	0.8248	1.5929	2.3894

## Data Availability

The data used to support the findings of this study can be made available by the corresponding author upon request.

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
