# Peer review of "Deep Learning-Based Near-Infrared Hyperspectral Imaging for Food Nutrition Estimation"

_foods, 2023, doi:10.3390/foods12173145_

Round 1
Reviewer 1 Report
I found your research very interesting and if it is successful in its application it would be potentially important. However, even when there is no information on another similar study, I believe that the discussion should include more references to the research that led to the conception of the present study, either the model or the use of near-infrared hyperspectral imaging.
Reviewer 2 Report
The authors correlated the protein content and the NIR spectra by using deep learning approach. The study appears to be very interesting. I think the authors can consider the amount of total carotenoids, both coming from the eggs (carotenoid,) and the tomatoes (lycopene). The carotenoids are also important in terms of nutritional value of a given food. They could look at the available data at different frequencies for other components, they need to take the wavelength accordingly. And the calculations would be very similar to what they have already done. Similarly, they could also evaluate the other components (carbohydrates, fats and water) and also can extent the results for matrix effects for the nutritional values of the components. The results can be correlated with the measurements and bioavailability/bioaccessibility evaluation of the food samples.
The authors should compare their results with the literature, which they did not.
The similarity report of the manuscript is 20 % without bibliographia. It is better to decrease it below 20%.

Reviewer 3 Report
Comments:
This paper is devoted to the feasibility of integrating deep learning with NIR-HSI for food nutrition estimation. The authors also proposed a novel deep learning method called OptmWave, which integrates two neural networks to simultaneously predict protein content and select wavelengths. This topic is interesting. However, before this paper is published, the following comments should be taken into account when revising the paper:
Major
1. Did “five-fold cross-validation” used in this study? How about results of ten-fold cross-validation?
2. Pros and cons of OptmWave should be specified.
3. How to access dataset used in this study? You should briefly introduce the link or access of dataset.
4. Why no comparison between traditional method and deep learning approach based on ROC curve? Such as logistic regression. Please give more illustrations.
5. The contribution of this paper should be highlighted.
6. The reference should be updated.
Reviewer 4 Report
Comments concerning manuscript foods-2537928
General comments:
The authors present a work about the feasibility of integrating deep learning with NIR-HSI for food nutrition estimation. The subject is interesting, but the manuscript quality is poor and a significant revision is required. To help the authors in the eventual revision of the manuscript, some pertinent comments are listed below:
- Do not write in the first person.
- Add a nomenclature or abbreviations section.
- See the authors' guide to correct the format of the references in the text.
- Add a logic diagram that outlines the work.
Keywords
- Do not capitalize keywords.
2.2.1. Hyperspectral imaging system
- Improve Figure 1.
2.3. Reference measurement of protein content
- Replace “using formula (2)” with “using the following equation”.
2.4. A deep learning framework for wavelength selection and regression simultaneously
- Add references.
2.4.3. Loss functions
- Add references.
2.5. Conventional approaches
- Add references.
3.4. Result analysis
- Compare with the results obtained by other authors.
Final assessment: Mayor Revision is required.
In the writing of the paper, the first person is mixed with the third person. Consider that in a scientific text it is better to use third person for its writing.
Round 2
Reviewer 3 Report
Thanks for your efforts on revision.
Reviewer 4 Report
The manuscript has greatly improved its quality and can be published.